# Are trajectories of depressive symptoms during the first half of drug-sensitive pulmonary tuberculosis treatment associated with loss to follow-up? A secondary analysis of longitudinal data

Paulo Ruiz-Grosso ![ORCID] ,[1,2] Christian Loret de Mola,[3] Larissa Otero,[1,4] Cesar Ugarte-Gil ![ORCID] [1,4]

For numbered affiliations see end of article.

**Correspondence to**
Dr Paulo Ruiz-Grosso;
paulo.ruiz@upch.pe

## ABSTRACT

**Objective** The objective of this study was to identify trajectories of depressive symptoms (DSs) during the first half of drug-sensitive pulmonary tuberculosis (PTB) treatment and examine their association with loss to follow-up (LTFU) in the second half.

**Design** This study involved a secondary analysis of longitudinal data to identify potential trajectories of DS and their relationship with LTFU.

**Setting** The study was conducted in first and second-level health centres located in San Juan de Lurigancho, Lima, Peru.

**Participants** Anonymised data from 265 individuals, including monthly measures of DSs from diagnosis to the completion of treatment, initiation of treatment for multidrug resistant TB, LTFU or death, were collected.

**Results** Three trajectories were identified: 'declining', 'growth' and 'high'. These trajectories were observed in 182 (68.7%), 53 (20%) and 30 (11.3%) of the 265 individuals, respectively, during the first half of PTB treatment. Compared with those with a 'declining' trajectory, individuals with a 'growth' trajectory had a higher likelihood of experiencing LTFU during the second half of PTB treatment, after controlling for sociodemographic factors and at least weekly alcohol use (OR 3.9; 95% CI 1.09 to 13.97, p=0.036).

**Conclusions** The findings suggest that a trajectory of increasing DSs during the first half of PTB treatment is associated with a higher risk of LTFU during the second half.

## INTRODUCTION

Tuberculosis (TB) remains the leading cause of death due to infectious diseases worldwide, resulting in 1.6 million deaths in 2021, with a disproportionate impact in low/middle-income countries. Following the onset of the COVID-19 pandemic, the detection of new TB cases declined, leading to an increase in TB mortality for the first time since 2015, making it the second-leading cause of death from infectious diseases globally. The emergence

## STRENGTHS AND LIMITATIONS OF THIS STUDY

⇒ We used monthly longitudinal measures of depressive symptoms starting from the time of diagnosis of drug-susceptible pulmonary tuberculosis.
⇒ The trajectories (patterns of change) of depressive symptoms were extracted from the data using latent class growth analysis.
⇒ The sample size was limited, and incidence of loss to follow-up was relatively low, which may impact the precision of the CIs.
⇒ A locally validated, five-item version of the CES-D scale was used to measure depression; however, there is no information available regarding the scale's reliability when used multiple times within a relatively short period.

of multidrug-resistant (MDR) strains poses a significant challenge to TB control, as 3.6% of cases worldwide require treatment for MDR or extensively drug-resistant TB (XDR). In Peru, 5.9% of cases initiating treatment between 2019 and 2022 were found to have MDR or XDR TB, placing the country at the 'higher-risk' category by the WHO.[1]

In Peru, treatment duration for drug-susceptible TB is at least 6 months, while MDR or XDR treatment extends up to 24 months. Failure or poor adherence during drug-susceptible TB treatment increases the risk of developing MDR or XDR strains, leading to more expensive, lengthier and uncomfortable treatment for patients.[2 3] Failure and loss to follow up (LTFU) are significant factors contributing to unsuccessful TB treatment, accounting for 10%–17% of drug-susceptible TB cases and 38% of MDR TB cases.[4 5] These outcomes have been associated with lower education levels, treatment for MDR TB, and comorbid depressive and substance use disorders.[5–9]

Depressive disorders are more prevalent among individuals undergoing TB treatment compared with the general population. In a previously published study among the same cohort, the prevalence of depression at baseline and at the end of TB treatment (after 6 months) was 38% and 14%, respectively, while population-based studies in Lima estimated the prevalence of depression to be close to 7%.[10 11] Among individuals treated for MDR-TB, 58% and 33% exhibited symptoms indicative of a major depressive episode at the start and end of treatment, respectively.[12] Depressive symptoms, including major depressive episodes, have been associated with increased risk of death (OR 2.85; 95% CI 1.52 to 5.36) and LTFU (OR 4.26; 95% CI 2.33 to 7.79) during TB treatment, suggesting that addressing these symptoms could be crucial for improving TB treatment outcomes.[13]

In this context, identifying clinically and epidemiologically relevant trajectories of depressive symptoms during the first half of pulmonary TB (PTB) treatment can help identify subgroups that may benefit from more robust and specialised interventions. These interventions can be tested and implemented to enhance clinical care and achieve public health goals. In this study, we analysed data from a cohort of patients undergoing drug-sensitive TB treatment to identify trajectories (patterns of change over time) of depressive symptoms associated with LTFU. This analysis aims to identify high-risk groups for these outcomes and explore opportunities for interventions.

## METHODS
### Study design and population
This analysis is based on a subsample of a cohort study designed to follow patients undergoing drug-sensitive PTB treatment in Lima, Peru. The original study was conducted at primary or secondary care public health centres and collected monthly information on depressive symptoms from the time of TB diagnosis until 6 months after the completion of TB treatment, death, LTFU or referral to another centre. The primary objective of the study was to investigate risk factors associated with negative outcomes during TB treatment, including death and treatment default. Individuals were invited to participate in the study before initiating TB treatment.

The study population consisted of individuals residing in San Juan de Lurigancho, a district located on the East side of Lima. This district is the most populous in Peru and is predominantly composed of migrants or descendants of migrants from the Andean regions of Peru.[14] The population in this area ranges from middle class to impoverished, and the district is one of the most affected with the burden of TB in Peru.[15]

Several exclusion criteria were applied in the study. Individuals younger than 18 years old, those with a previous diagnosis of TB, pregnant women, illiterate individuals and those with any mental or physical condition that could hinder the informed consent process were excluded.

For this analysis, we included information on individuals that had at least four complete measurements of depressive symptoms (baseline and 3 monthly follow-ups), and excluded those who were diagnosed with MDR-TB at any point during follow-up and those who died during the first 3 months of treatment and thus were not at risk during the second half of TB treatment and could not contribute to depressive measures in the first half of TB treatment. As a result, data from 265 individuals were included in the data analysis.

### Study variables
The main outcome variable was LTFU during the second half of TB treatment defined as the interruption of anti-TB treatment for at least 30 days without medical indication, this information was obtained from the official clinical records as part of the original study.

The main predictor variable was trajectories (patterns of change over time) of depressive symptoms during the first half of PTB treatment. These trajectories were identified as part of the data analysis of this study using the score of a five-item, locally validated version of the Center for Epidemiological Studies Depression Scale in Spanish.[16] Participants completed this scale at the beginning of TB treatment and then monthly thereafter until treatment completion (6 months), LTFU or the initiation of treatment for MDR-TB. Each participant contributed between 1 and 7 monthly scores of depressive symptoms.

In addition, several covariates were measured at the start of follow-up, including age, sex, marital status, education, employment status and frequency of alcohol use.

### Statistical analysis
The data analysis was conducted in two steps. First, latent class growth analysis was employed to identify significant variations in the pattern of depressive symptom scores during the first 3 months of follow-up.[17] Monthly visits were used as the time scale variable, and the 'LISTWISE' option was used to handle missing data, which were designated accordingly in the database. The initial stage of the analysis involved 500 random starts, followed by 10 optimisations in the final stage.

Linear and quadratic models were explored for the trajectories of depressive symptoms. Linear models allowed for simpler models without changes in the direction of the trajectory, while quadratic models allowed for the exploration of possible changes in the trajectory direction for better understanding. No covariates were included in this part of the analysis. The code for the selected model analysis can be found in online supplemental material 1.

Next, various fit indices, including entropy, root mean squared error of approximation, Comparative Fit Index, Akaike information criteria, Bayesian information criteria, Vuong-Lo-Mendell-Rubin and Bootstrapped likelihood ratio tests, were used to select the model that best fit the data. This step provided information on the

trajectories (intercept, linear and quadratic terms) and assigned each participant to a specific trajectory.

In the second step, logistic regression was employed to test the hypothesis that individuals with worse trajectories of depressive symptoms during the first half of PTB treatment were more likely to experience LTFU during the second half of TB treatment compared with those with more favourable trajectories. Multiple logistic regression was used to adjust for potential confounders, including sex, age, marital status, education and employment status. Only observations with complete information on these covariates were included in this analysis.

The role of at least weekly alcohol use at the start of follow-up (defined as consuming alcoholic beverages at least once per week) was explored as a potential confounder by adjusting for it and as an effect modifier using stratification.

### Patient and public involvement
None.

### RESULTS
Of the initial 344 participants, 34 were excluded due to MDR-TB, and 45 were excluded due to missing data on the depression score during the first half of drug-sensitive PTB treatment. The final analysis included data from 265 participants. Among these participants, 13 (4.9%) were lost to follow-up during the second half of TB treatment.

The mean age of the participants was 29.6 (SD=12.9), with 3 missing values. Of the analysed participants, 145 (54.7%) were male, with 3 missing values, and the majority were single (155, 58%), with 4 missing values. Among the participants, 112 (42.3%) had completed secondary education (high school), with 4 missing values, while 192 (72.5%) were unemployed at the start of TB treatment, also with 4 missing values. Further details about the sample can be found in table 1.

During the baseline evaluation, 120 individuals (38.7%) exhibited symptoms suggesting major depressive episodes. This number decreased to 33 (13.5%) in the last evaluation, which occurred at the end of TB treatment. Additional information on this topic is reported in more detail elsewhere.[13] The overall incidence of symptoms suggesting a major depressive episode after the initiation of TB treatment was observed in 85 individuals (30.3%).

### Identification of trajectories for depressive symptoms during the first half of PTB treatment
Two quadratic models with three and four identified trajectories demonstrated acceptable goodness-of-fit parameters (refer to table 2). The first model identified three trajectories: (1) 'decline over time of depressive symptoms' (69% of participants), (2) 'growth over time' (20%) and (3) 'high initial score with a decline over time' (11%). The second model identified four trajectories: (1) 'decline over time of

**Table 1** Sample characteristics of individuals undergoing drug-susceptible TB treatment in Lima, Peru (n=265)

| | | N (%) |
|---|---|---|
| Male sex | | 145 (54.7) |
| Age at enrolment* | | 29.6 (12.9) |
| Marital status | | |
| | Single | 155 (58.5) |
| | Married or cohabitant | 90 (34.0) |
| | Divorced or widowed | 20 (7.5) |
| Attained education | | |
| | Incomplete elementary | 20 (7.5) |
| | Complete elementary | 75 (28.3) |
| | Complete secondary | 112 (42.3) |
| | Superior education | 58 (21.9) |
| Currently employed | | 73 (27.55) |
| Negative outcomes during the second half of TB treatment | | |
| | Death | 0 (0.0%) |
| | Loss to follow-up | 13 (4.91) |
| | Total | 13 (4.91) |

*Mean.
TB, tuberculosis.

depressive symptoms' (65% of participants), (2) 'slow growth over time' (21%), (3) 'rapid growth over time' (5%) and (4) 'high initial score with decline over time' (9%). Both models significantly differed from a solution with one less trajectory, as indicated by the Vuong-Lo-Mendell-Rubin test (p=0.035 and 0.001 for the three and four trajectory models, respectively) and the Bootstrapped likelihood ratio test (p<0.001 for both). Additional details on the goodness-of-fit estimates and estimated parameters can be found in online supplemental material 2, while graphical representations of the trajectories are provided in figures 1 and 2, online supplemental figure A.

For further analysis, we selected the thee-trajectory solution because the four-trajectory model included two trajectories that accounted for less than 10% of the sample, and the two growth trajectories were quite similar. Thus, the three-trajectory model was deemed more parsimonious.

### Association between trajectories of depressive symptoms on the first half of PTB and negative outcomes during the second half of PTB treatment
Compared with individuals with a 'decline over time of depressive symptoms' trajectory, those with a 'growth over time' during the first half of PTB treatment had odds of LTFU during the second half that were 3.74 times higher (95% CI 1.15 to 12.14). This association remained significant even after controlling for age, sex, at least weekly use of alcohol, marital status, education and job status (OR 3.90; 95% CI 1.09 to

**Table 2** Parameters of estimated trajectories of depressive symptoms during the first half of drug susceptible PTB treatment

| | 1 trajectory | | 2 trajectories | | 3 trajectories | | 4 trajectories | |
|---|---|---|---|---|---|---|---|---|
| | Linear | Quadratic | Linear | Quadratic | Linear | Quadratic | Linear | Quadratic |
| **Trajectory 1** | | | | | | | | |
| % of class | 1.00 | 1 | 0.18 | 0.85 | 0.08 | 0.12 | 0.05 | 0.21 |
| Intercept | 4.40 | 4.71 | 9.40* | 3.87 | 4.70* | 10.24* | 3.92* | 5.25* |
| Slope | −0.655 | −1.39 | −1.74* | −1.56 | 1.24* | −0.6 | 2.15* | −3.53* |
| Quadratic term | NA | 0.229 | NA | 0.37 | NA | −0.7 | NA | 1.224* |
| **Trajectory 2** | | | | | | | | |
| % of class | NA | NA | 0.82 | 0.15 | 0.79 | 0.2 | 0.22 | 0.09 |
| Intercept | NA | NA | 3.33 | 9.38 | 3.4 | 5.8 | 3.65* | 10.67* |
| Slope | NA | NA | −0.44* | −0.42 | −0.50 | −4.00* | 0.58** | −0.76 |
| Quadratic term | NA | NA | NA | −0.56 | NA | 1.50* | NA | −0.71 |
| **Trajectory 3** | | | | | | | | |
| % of class | NA | NA | NA | NA | 0.13 | 0.69 | 0.09 | 0.05 |
| Intercept | NA | NA | NA | NA | 10.87* | 3.46* | 11.97* | 5.97* |
| Slope | NA | NA | NA | NA | −2.77* | −0.77* | −3.34* | −2.82 |
| Quadratic term | NA | NA | NA | NA | NA | ~0 | NA | 1.46* |
| **Trajectory 4** | | | | | | | | |
| % of class | NA | NA | NA | NA | NA | NA | 0.64 | 0.65 |
| Intercept | NA | NA | NA | NA | NA | NA | 3.57* | 3.48 |
| Slope | NA | NA | NA | NA | NA | NA | −0.84* | −0.67* |
| Quadratic term | NA | NA | NA | NA | NA | NA | NA | −0.05* |

*P<0.05.
NA, not available.

13.97; p=0.036). A small but statistically non-significant association with LTFU was found for individuals with a 'high initial score with decline over time' trajectory when compared with those with a 'decline over time of depressive symptoms' (OR 1.28; 95% CI 0.13 to 12.54; p=0.829). Details of the bivariate analyses, including coefficient regression of potential confounders, can be found in table 3 for the consideration of readers.

## Alcohol use in the association between trajectories of depressive symptoms in the first half of PTB and negative outcomes during the second half of PTB treatment

Alcohol use at least once per week at the beginning of anti-TB treatment was associated with a higher risk of LTFU during the second half of TB treatment (OR 2.52; 95% CI 1.01 to 6.28; p=0.046). However, this association lost significance when adjusting for trajectories of depressive symptoms, sex, age, marital status, education level and job status (see table 3).

On stratification, we found that the association between having an increasing trajectory of depressive symptoms during the first half of PTB treatment (compared with those with a decreasing trajectory) and LTFU in the second half remained significant only in the presence of at least weekly use of alcohol (OR 13.3; 95% CI 1.32 to 134.92; p=0.028).

## Profiling of missing data

During the first half of PTB treatment 45 (14.5%) did not complete at least one of the required four depression measurements and were excluded from the study. In this group, 21 (50%) individuals were female, with a mean age of 33.1, while the proportions of married, single and divorced individuals were 40%, 37.8% and 13.3%. The total of deaths during follow-up occurred during the first half of PTB treatment. Of the 20 events of LTFU, 13/265 (4.9%) happened in the group that was analysed, and 7/45 (15.5%) in the group of individuals that were excluded from the analysis due to incomplete data on depressive symptoms during the first half of PTB treatment.

## DISCUSSION

Our findings suggest that an increase in depressive symptoms during the first half of drug susceptible PTB treatment is associated with a higher likelihood of LTFU during the second half, in comparison to patients whose depressive symptoms decreased during this period. We identified three trajectories of depressive symptoms during TB treatment: decreasing, increasing and high depressive symptoms with a decrease over time.

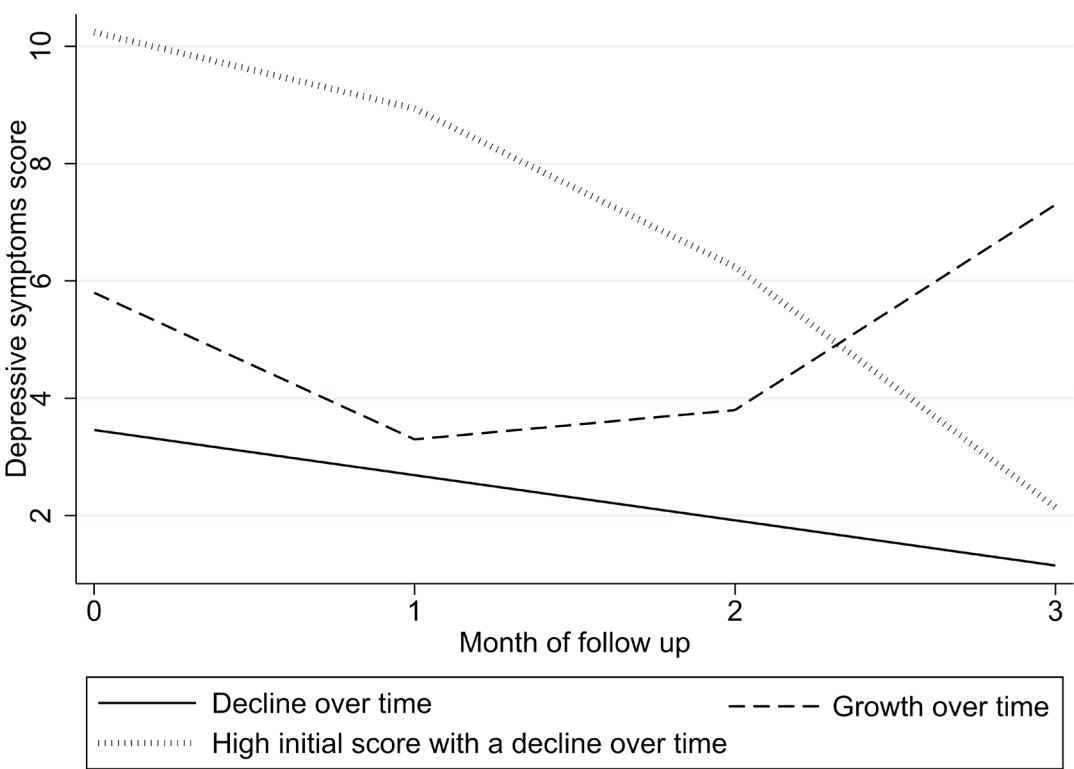

**Figure 1** Trajectories of depressive symptoms on the first half of PTB treatment, three trajectories model. PTB, pulmonary tuberculosis.

Furthermore, we found that drinking alcohol at least weekly at the start of TB treatment may modify the association between depressive symptoms trajectories and LTFU. Unfortunately, no analysis of deaths could be conducted as all deaths occurred during the first half of PTB treatment.

These findings have important implications for both researchers and clinicians, as they highlight characteristics that may indicate a higher risk of LTFU in the later stages of TB treatment that might be identified early. This knowledge can contribute to the development of targeted interventions that address mental health disorders, such as depressive and drug use disorders, and ultimately improve TB care.

To the best of our knowledge, this is the first study to investigate the trajectories of depressive symptoms during TB treatment and their association with LTFU during treatment. Previous research has examined the patterns of depressive symptoms in other contexts. For instance, Owora identified four trajectories of depressive symptoms during HIV treatment: low symptoms, high symptoms, episodic symptoms with temporary increases, and a slow increase over time. These trajectories bear similarities to the ones found in our study, with the declining trajectory in our study corresponding to low symptoms over time.

The main difference lies in the presence of at least moderately severe symptoms at the beginning of our follow-up, which subsequently decreased and remained low throughout the study period. This discrepancy may be attributed to our study capturing the first measurement shortly after TB diagnosis, whereas Owora 's study initiated follow-up at the first clinic visit, which may not coincide with the diagnosis.

Owora also identified two trajectories of increasing depressive symptoms, accounting for approximately 9% of participants: an episodic trajectory with symptom increases between the second and third year of follow-up, and a sustained growth trajectory over the 6-year follow-up. In comparison, our study revealed a higher proportion (>20%) of participants in this group who initially experienced a decline in symptoms before they started to increase in magnitude. The disparity in follow-up duration (months vs years) may explain these differences.

Both studies highlight the influence of depressive symptom trajectories on treatment outcomes. For instance, compared with the group with consistently low symptoms, those with an ascending trajectory had a 50% higher likelihood of a lower CD4 count during follow-up, which aligns with our research demonstrating a higher risk of LTFU.[18]

The identified trajectories also resemble those observed in studies investigating the trajectory of depressive symptoms following the loss of a spouse. Grief-related studies, such as those by Maccallum *et al* and Galatzer-Levy *et al.*, demonstrated trajectories characterised by both high and low scores, as well as patterns of decline and growth.

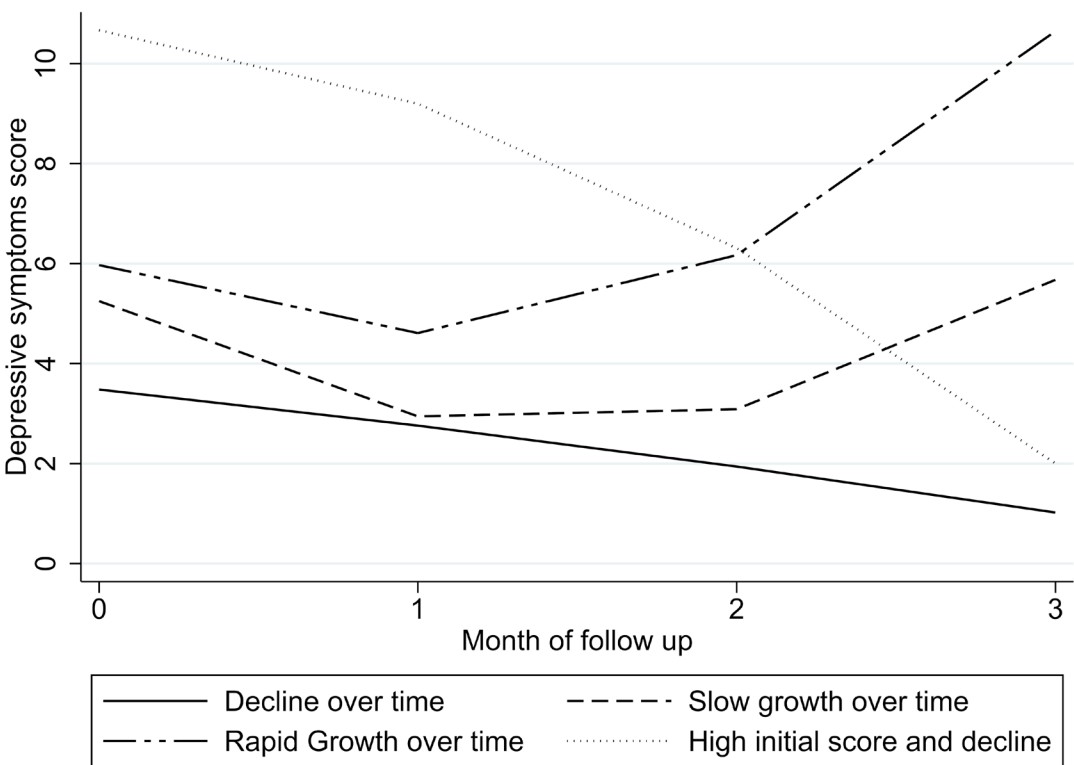

**Figure 2** Trajectories of depressive symptoms on the first half of PTB treatment, four trajectories model. PTB, pulmonary tuberculosis.

**Table 3** Variables associated with negative outcomes of TB during the second half of drug-susceptible TB treatment

| | Simple regression | | | Multiple regression | | |
|---|---|---|---|---|---|---|
| **Variables** | **OR** | **95% CI** | **P value** | **OR** | **95% CI** | **P value** |
| Trajectories of depressive symptoms during the first 3 months of drug-susceptible TB treatment | | | | | | |
| Decline over time | Ref. | – | – | Ref. | – | – |
| Growth over time | 3.74 | 1.15 to 12.14 | 0.028 | 3.9 | 1.09 to 13.97 | 0.036 |
| High initial score with decline over time | 1.01 | 0.18 to 8.71 | 0.992 | 1.28 | 0.13 to 12.54 | 0.829 |
| Frequent alcohol use | 2.52 | 1.01 to 6.3 | 0.046 | 3.18 | 0.87 to 11.62 | 0.08 |
| Male sex | 1.35 | 0.54 to 3.22 | 0.498 | 2.21 | 0.50 to 9.75 | 0.296 |
| Age (years) | 1.01 | 0.97 to 1.04 | 0.742 | 0.98 | 0.92 to 1.05 | 0.653 |
| Marital status | | | | | | |
| Single | Ref. | – | – | Ref. | – | – |
| Married or cohabitant | 1.35 | 0.54 to 3.38 | 0.518 | 1.86 | 0.43 to 8.09 | 0.405 |
| Divorced or widowed | 1.92 | 0.50 to 7.40 | 0.343 | 2.35 | 0.33 to 16.70 | 0.392 |
| Attained education | | | | | | |
| Incomplete elementary | Ref. | – | – | Ref. | – | – |
| Complete elementary | 0.78 | 0.15 to 4.11 | 0.766 | 0.97 | 0.07 to 13.05 | 0.983 |
| Complete secondary | 0.95 | 0.19 to 4.63 | 0.948 | 1.47 | 0.11 to 18.74 | 0.768 |
| Superior education | 0.92 | 0.17 to 5.07 | 0.921 | 2.41 | 0.15 to 39.50 | 0.538 |
| Currently employed | 0.69 | 0.25 to 1.93 | 0.482 | 0.96 | 0.26 to 3.56 | 0.956 |

TB, tuberculosis.

However, we did not observe a trajectory of consistently low depressive symptoms. Our study's follow-up commenced shortly after diagnosis and TB treatment initiation, corresponding to a higher frequency of depressive symptoms possibly related to mourning. As the mourning associated with the loss of health and the diagnosis of a stigmatised disease such as TB may resolve with antibiotic treatment, exposure and improved information about the disease prognosis, we anticipate fewer individuals will experience a sustained depressive state compared with mourning following the death of a loved one.[10]

Weekly alcohol use was identified as a significant risk factor associated with a twofold increase in the likelihood of LTFU, as reported by Lackey et al.[19] Alcohol use has consistently been recognised as a risk factor for both TB disease and poor treatment adherence, although variations in its definition may contribute to differences in significance when controlling for illegal drug use.[6 7 20–22] The interaction between alcohol use disorders and depressive syndromes is complex, and the presence of one may increase the likelihood of the other. However, our findings demonstrate that an increasing trajectory of depressive symptoms is only significantly associated with a higher chance of negative outcomes in the subgroup of individuals who engage in at least weekly alcohol consumption.

As important limitations of this study, it should be noted that the instrument used to measure depressive symptoms, a five-item version of the CES-D scale, was validated in a similar population and demonstrated good sensitivity and specificity (>90%). However, its performance has not been locally evaluated when multiple measurements are taken over a relatively short period of time (6 months). Additionally, no confirmatory assessment of depression, such as a clinician interview or measurement of impairment, was conducted. The measured depressive symptoms using the short version of the CES-D scale may be influenced by the burden of stigma associated with being diagnosed with a disease such as TB. Furthermore, there is no information available regarding whether participants received any therapeutic intervention for depression during TB treatment.

Regarding the study population, it is important to note that the results can only be extrapolated to individuals receiving treatment for drug-sensitive TB in primary care settings in urban and suburban areas, within the context of a government programme specifically designed for identifying and treating TB in Peru. The sample size of the study may not have been ideal for precisely defining trajectories of depressive symptoms when small proportions were observed. For instance, for the 'high initial score with a decline over time' trajectory, which was assigned to 11% of individuals, the sample size was not sufficient to conclusively determine if there is evidence of a change in the trajectory over time or if individuals with this trajectory have a higher likelihood of LTFU compared with those with a decreasing trajectory.

Readers should also be aware of the possibility of selection bias influencing the results, as LTFU and possibly the trajectories of depressive disorders may be associated with a higher likelihood of missing data. Finally, caution should be exercised when interpreting adjusted coefficients of the multiple logistic regression model, as even though the coefficient of the main variable may be properly adjusted, the coefficients of covariates may not be, as discussed by Westreich and Greenland and exemplified by Bandoli et al.[23 24]

## CONCLUSION

At least three trajectories (decline, growth and high initial score with decline over time) of depressive symptoms may be observed in individuals during the first half of drug susceptible PTB treatment. Those with increasing symptoms may be at a heightened risk of LTFU in the final 3 months of treatment. The presence of alcohol use at the start of treatment may modify this association. Further studies with greater statistical power are necessary to enhance the accuracy of our estimates, as well as to validate and broaden these findings.

**Author affiliations**
[1]Facultad de Medicina Alberto Hurtado, Universidad Peruana Cayetano Heredia, San Martín de Porres, Lima, Peru
[2]Programa de Control de Tuberculosis, Dirección de Redes Integradas de Salud Lima Norte, Estado Peruano Ministerio de Salud, Lima, Peru
[3]GPIS Grupo de Pesquisa e Inovação em Saúde, Universidade Federal do Rio Grande, Rio Grande, Brazil
[4]Instituto de Medicina Tropical Alexander von Humboldt, Universidad Peruana Cayetano Heredia, San Martín de Porres, Peru

**Acknowledgements** The authors would like to acknowledge the field team of the Tuberculosis Research Unit at the Instituto de Medicina Tropical Alexander von Humboldt of the Universidad Peruana Cayetano Heredia.

**Contributors** PR-G was responsible for conceptualisation, analysis, interpretation, writing the original draft, review and editing. CLdM was responsible for analysis, interpretation and critical review. LO was responsible for data acquisition, interpretation and critical review. CU-G was responsible for conceptualisation, supervision and critical review. CU-G is responsible for the overall content as the guarantor.

**Funding** The original study was funded by the Peru ICOHRTA Network for AIDS/TB Research Training (NIH Grant 1U2RTW007368-01A1- Fogarty International Center, Lima, Peru) (CU-G), with partial support from the Belgian Cooperation through a project of institutional collaboration between the Institute of Tropical Medicine in Antwerp, Belgium, and the Instituto de Medicina Tropical Alexander von Humboldt in Lima, Peru (LO). LO is supported by an Emerging Global Leader Award (K43TW011137) from the Fogarty International Center at the National Institutes of Health. PR received the CIENCIACTIVA (Consejo Nacional de Ciencia, Tecnología e Innovación Tecnológica) scholarship for the Doctorate Program in Epidemiological Research Sciences at the Facultad de Salud Pública y Administración, Universidad Peruana Cayetano Heredia.

**Disclaimer** The funders had no role in the study design, data collection, analysis, the decision to publish, or the preparation of the manuscript.

**Competing interests** None declared.

**Patient and public involvement** Patients and/or the public were not involved in the design, or conduct, or reporting, or dissemination plans of this research.

**Patient consent for publication** Consent obtained directly from patient(s).

**Ethics approval** This study involves human participants and ethical approval for this study was obtained from the Universidad Peruana Cayetano Heredia IRB, under

review number (SIDISI number) 203884. All participants participated in the informed consent process and provided informed consent prior to enrollment in the study. For the purposes of this study, the data were received without any information that could identify the participants. Treating physicians were informed of the depression status of each participant.

**Provenance and peer review** Not commissioned; externally peer reviewed.

**Data availability statement** Data are available on reasonable request.

**Open access** This is an open access article distributed in accordance with the Creative Commons Attribution 4.0 Unported (CC BY 4.0) license, which permits others to copy, redistribute, remix, transform and build upon this work for any purpose, provided the original work is properly cited, a link to the licence is given, and indication of whether changes were made. See: https://creativecommons.org/licenses/by/4.0/.

**ORCID iDs**
Paulo Ruiz-Grosso http://orcid.org/0000-0001-8003-5171
Cesar Ugarte-Gil http://orcid.org/0000-0002-2833-9087

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
