## [Reviewer comments · BMJ Open]

This paper was submitted to a another journal from BMJ but declined for publication following peer review. The authors addressed the reviewers' comments and submitted the revised paper to BMJ Open. The paper was subsequently accepted for publication at BMJ Open.

ARTICLE DETAILS

TITLE (PROVISIONAL)	Are trajectories of depressive symptoms during the first half of drug-sensitive pulmonary tuberculosis treatment associated with loss to follow-up? A secondary analysis of longitudinal data
AUTHORS	Ruiz-Grosso, Paulo; Loret de Mola, Christian; Otero, Larissa; Ugarte-Gil, Cesar

VERSION 1 – REVIEW

REVIEWER	Loveday, Marian South African Medical Research Council
REVIEW RETURNED	24-Nov-2022

GENERAL COMMENTS	Thank you for asking me to review this manuscript in which the authors report the effect of depressive symptoms in the first half of TB treatment on loss to follow up in the second half of treatment. The authors are addressing an important and topical issue in this manuscript, as although depression and its effect on retention in care has been well documented in HIV and other disease literature, it has not been that well researched or reported on in the TB literature. Most of my comments relate to the readability of this manuscript. As the manuscript is addressing an important issue, it is important that it is accessible and can be easily read. To this end, revision by an English language editor is needed. For example, in the first sentence of the introduction 'Tuberculosis was the first cause of death...', this is not grammatically correct English and so the manuscript is not easy to read. And instead of using terms such as 'Full sensitive' or 'fully sensitive' TB, I would suggest using the accepted terminology 'drug susceptible' TB rather which can then be abbreviated to DS-TB. In addition, to ensure the manuscript can be read by those who work in TB but are not that familiar with TB, a number of terms need to be defined in the methodology section. These include "Decline over time of depressive symptoms", "Growth over time", "Slow growth over time", "High growth over time" and 'quadratic solutions'. In the discussion section, the manuscript should be written in ordinary English and not in research terminology:
--

Line 188 – 192: This paragraph should be written in ordinary English and not in research terminology, highlighting your main study finding that an increase in depressive symptoms was associated with LTFU in the second half of treatment.

Lines 209 – 213: 'High initial score with a decline over time' trajectory which was estimated for 11% of individuals, neither the slope nor the quadratic term showed a rejection of the null hypotheses, thus limiting our understanding of the behavior over time of the individuals that started receiving TB treatment with high levels of depressive symptoms.' Again, this sentence should be written in ordinary English and not in research terminology, focus on what your finding was.

Lines 219 – 220: 'four trajectories for depressive symptoms low and high chronic symptoms, episodic and slow rise.' Again, for someone who has not read this article, this sentence is not understandable.

Additional comments

Lines 58 – 60: Update with data from the recently released 2022 Global TB report.

In the methods section it is not clear how study participants were selected or followed up to determine if they completed treatment or were lost to follow up.

Line 75 – 78: The article referenced here does not compare the proportion of TB patients with a major depressive episode with the general population in Peru.

Line 156 – 158: The type of statistical analysis used should be in the methodology section, not the results section.

Lines 162 – 165: I think this section should also be in the methodology section and table 2 should be a supplementary table.

Line 194-195: 'As all deaths occurred during the FH of treatment no analysis could be done of this outcome.' As I understand this sentence those who died during the first 3 months of treatment were not included in the analysis. This should be included in your methodology, not in the discussion.

Lines 196 – 215: These are your study limitations and should be at the end of your discussion section.

Line 224: 'follow up': Do you mean monthly follow up during TB treatment?

References: Unnecessay detail in the references must be removed. For example:

Muture BN, Keraka MN, Kimuu PK, Kabiru EW, Ombeka VO, Oguya F. Factors associated with default from treatment among tuberculosis patients in Nairobi province, Kenya: a case control study. BMC Public Health [Internet]. 2011 Sep 9 321 [cited 2019 Jun 26];11(1):696. Available from: 322

<http://bmcpublichealth.biomedcentral.com/articles/10.1186/1471-2458-11-696>

	Please check on the author guidelines page whether all detail in the last two lines above is necessary.
--	---

REVIEWER	Sweetland, Annika Columbia University Vagelos College of Physicians and Surgeons, Psychiatry
REVIEW RETURNED	13-Dec-2022

GENERAL COMMENTS	Major revision This manuscript makes a unique contribution the literature in that it is the first to examine depression trajectories in TB patients in the absence of mental health treatment. While I am not qualified to assess the statistical methods, if valid, the findings are of potential interest and use in the field. The authors describe several depressive symptom trajectories (declining, growth or high), and effectively pinpoint one trajectory as being the most strongly associated with LTFU ("growth"). If true, this seems to suggest that ongoing monitoring of depressive symptoms in TB patients to identify those whose symptoms increase after baseline may help to identify this high risk group for targeted intervention. However, the authors do not explore this clinical implication. The baseline prevalence of depressive symptoms is also unclear, making the entire picture difficult to interpret. In order to strengthen this paper, I would want to have a clearer picture of the following:  1) what proportion of patients met criteria for "probable depression" at baseline (i.e. prevalence)? 2) what proportion of patients developed probable depression during the course of treatment (incidence)? 3) Of those with baseline depression, what proportion remitted without mental health treatment? What does this mean for clinical management? 4) Did all individuals in the "growth" trajectory meet criteria for probable depression at baseline? 5) How do you interpret that "high" was not as strong a predictor of LTFU as "growth" (assuming I am understanding this finding correctly)? 6) Overall, what are the clinical implications of these findings regarding TB and depression service integration? Biostatistician review methods. Minor: sample size should be reported in abstract and methods section; fewer abbreviations would make the paper easier to read
--

REVIEWER	Kabali, Conrad Health Quality Ontario
REVIEW RETURNED	05-Feb-2023

GENERAL COMMENTS	I kindly suggest revising the terminology in page 12, line 35 to "95% Confidence Interval (CI)" instead of "IC95%". On line 42, it would be beneficial to consider the magnitude of the point estimate (OR=1.28) in addition to the tests of significance when interpreting the results. In regards to Table 3, it may be appropriate to reconsider the inference made on covariates that serve only as potential confounders. For instance, if evaluating the effect of alcohol, it may be necessary to remove covariates that are in the causal pathway. Finally, on page 14, line 5, it is recommended to focus on the effect size rather than just the significance level, as the point
--

	estimate may not be precise enough to form a conclusion. Further data collection may be needed to make a conclusion.
--	--

REVIEWER	Talbot, Denis Université Laval
REVIEW RETURNED	12-Feb-2023

GENERAL COMMENTS	This is a statistical review of the paper entitled “Trajectories of depressive symptoms during the first half of drug sensitive tuberculosis treatment and loss to follow up in Lima, Peru”. This paper is concerned with estimating the association between the trajectories of depressive symptoms during the first half of tuberculosis treatment with the probability of being “lost to follow up” during the second half of the treatment. I found the paper to read well and to provide a good amount of detail. I suggest some relatively minor revisions to both methods and reporting. While I focus on statistical aspects of the work in my paper, I wonder about the relevance of this study considering the small sample size and the ensuing imprecision of statistical results - which the authors appropriately recognize as a limitation of their study. Specific comments/questions/suggestions:  - Abstract: The problem under study wasn't very clear to me when reading the abstract. It wasn't clear to me that “loss to follow-up” was relative to patients' treatment and not relative to their participation in a longitudinal study. While I appreciate the challenges of space constraint, I recommend the authors revise the abstract to clarify. - Introduction:  o What is XDR-TB? o I think the authors could strengthen the rationale for their study. What is the importance of knowing the association between depressive symptom trajectories and loss to follow-up? Can this be used by clinicians or public health to prevent loss to follow-up? Is this information relevant for future research? - Methods:  o The exclusion of individuals with less than four measurements of depressive symptoms and those who developed MDR-TB during the follow-up may induce selection bias. If deemed possible and relevant, the authors may consider controlling this potential selection bias using inverse probability of censoring weighting or multiple imputation; the former being more easily applied than the latter with latent growth class analysis. o CFI should be comparative fit index instead of comparative fix index. o The authors may consider using the Guidelines for Reporting Trajectory Studies (GRoLTS) in addition to the STROBE to ensure that all appropriate information for this type of analysis is reported transparently. Some of the missing information for the methods include: Number of random start values, total number of models fitted including a one-class solution. o Please clarify when the covariates that were used for adjustment measured. o Were there any missing data on the covariates? If yes, please describe in the results these missing data and clarify in the methods how were they treated. o Please provide the code for performing the trajectory analysis as supplementary material as recommended by GRoLTS.
---

	- Results:  o I suggest adding a comparison of subjects included with those excluded to better inform on the potential for selection bias due to these exclusions. o Please remove the trajectories from table 1 since it is the result of an analysis and not a descriptive variable that can be directly measured. o Please remove the IC 95% from the legend of table 1 since this abbreviation is not used in the table. o I suggest revising the interpretation of non-significant results as being inconclusive rather than as an absence of association. With very large confidence intervals, the data really provide no meaningful information on presence, absence or strength of a potential association. o If the covariates are considered as potential confounders, then their coefficients should not be interpreted and their reporting in Table 2 may be misleading (subject to Table 2 fallacy, see https://doi.org/10.1093/aje/kws412). o Please add a plot of the observed and predicted means for each trajectory group to visually assess the goodness of fit (see GRoLTS). - Discussion:  o I find the paper to insufficiently describe the practical implications of their results. Given the lack of statistical power, perhaps there are no immediate implications? In my opinion, minor English language revision is needed.
--	--

VERSION 1 – AUTHOR RESPONSE

Reviewer: 1

Dr. Marian Loveday, South African Medical Research Council

Comments to the Author:

Thank you for asking me to review this manuscript in which the authors report the effect of depressive symptoms in the first half of TB treatment on loss to follow up in the second half of treatment. The authors are addressing an important and topical issue in this manuscript, as although depression and its effect on retention in care have been well documented in HIV and other disease literature, it has not been that well researched or reported on in the TB literature.

1. Most of my comments relate to the readability of this manuscript. As the manuscript is addressing an important issue, it is important that it is accessible and can be easily read. To this end, revision by an English language editor is needed. For example, in the first sentence of the introduction 'Tuberculosis was the first cause of death...', this is not grammatically correct English and so the manuscript is not easy to read.

And instead of using terms such as 'Full sensitive' or 'fully sensitive' TB, I would suggest using the accepted terminology 'drug susceptible' TB rather which can then be abbreviated to DS-TB.

We have performed a complete review of the new version of the manuscript in order to improve its readability and use of English. We also adequate the use of terms according to the reviewer's suggestions.

2. In addition, to ensure the manuscript can be read by those who work in TB but are not that familiar with TB, a number of terms need to be defined in the methodology section. These include "Decline

over time of depressive symptoms”, “Growth over time”, “Slow growth over time”, “High growth over time” and ‘quadratic solutions’.

In the Methods section when linear and quadratic methods are first described, we have added a small explanation of what it means: “We explored linear and quadratic models for trajectories, as linear models would allow us simpler models without any change in the direction of trajectory, and quadratic models allow us to explore possible changes in the direction of the trajectory that might allow better understanding”.

3. In the discussion section, the manuscript should be written in ordinary English and not in research terminology:

Line 188 – 192: This paragraph should be written in ordinary English and not in research terminology, highlighting your main study finding that an increase in depressive symptoms was associated with LTFU in the second half of treatment.

We have redacted the information in this paragraph in order to highlight our main results in less technical terms.

Lines 209 – 213: ‘High initial score with a decline over time’ trajectory which was estimated for 11% of individuals, neither the slope nor the quadratic term showed a rejection of the null hypotheses, thus limiting our understanding of the behavior over time of the individuals that started receiving TB treatment with high levels of depressive symptoms.’ Again, this sentence should be written in ordinary English and not in research terminology, focus on what your finding was.

We have changed the paragraph in favor of better readability.

Lines 219 – 220: ‘Four trajectories for depressive symptoms low and high chronic symptoms, episodic and slow rise.’ Again, for someone who has not read this article, this sentence is not understandable.

We have changed the phrasing of this paragraph in order to make it easier to understand out of context. It reads now: “found four trajectories for depressive symptoms during HIV treatment: two of these showed low and high symptoms across the period of the study, another one showed episodic symptoms, meaning increases during a period and the other a slow rise of depressive symptoms over time. These trajectories were similar to those found in this study, in which the higher proportion of the population which showed a declining trajectory on our study would correspond to the low symptoms over time”.

Additional comments

- Lines 58 – 60: Update with data from the recently released 2022 Global TB report. Data is now updated

- In the methods section, it is unclear how study participants were selected or followed up to determine if they completed treatment or were lost to follow-up.
 - The follow-up information was taken during the original study execution from the original clinical records, while participants were selected at the start or just before the start of TB treatment. This information is now added to the relevant section.

- Line 75 – 78: The article referenced here does not compare the proportion of TB patients with a major depressive episode with the general population in Peru.

○ The referenced article does not directly compare the prevalence of TB and the general population, rather we used two articles to make this indirect comparison, but we forgot to cite de second, which is a population-based study in Lima. We have now added the second reference.

● Line 156 – 158: The type of statistical analysis used should be in the methodology section, not the results section.

○ In the results section, we included all the steps taken that involve the use of data, which we believe should be in this section because it allows the reader to be presented with a better understanding of how some of the results lead to the selection of models than in turn allows for further analysis. For example, we include the process of identification of the trajectories, which shows that at least two models were adequate and then we justify why we chose one over the other for further steps in the analysis.

○ We think that all the information a reader might need in order to replicate the study from the same starting point as us is contained in the methods section.

● Lines 162 – 165: I think this section should also be in the methodology section and table 2 should be a supplementary table.

○ We considered this possibility, however, we think that in order to reach the conclusion on what model to choose, the first part of the analysis needs to be performed, so it is not a procedure that might be planned. \

○ We think the information in Table 2 informs on the characteristics of the models that were explored so the reader is allowed to compare them. We are open to using it as a supplementary table if the editors consider it necessary, however.

● Line 194-195: 'As all deaths occurred during the FH of treatment no analysis could be done of this outcome.' As I understand this sentence those who died during the first 3 months of treatment were not included in the analysis. This should be included in your methodology, not in the discussion.

○ We included this information was included in the methodology section.

● Lines 196 – 215: These are your study limitations and should be at the end of your discussion section.

○ We have moved the limitations to the end of the discussion section, as suggested by the reviewer.

● Line 224: 'follow up': Do you mean monthly follow-up during TB treatment?

○ We have worded it: "...remained low over the duration of follow up", in order to improve the clarity of statement.

● References: Unnecessary detail in the references must be removed. For example:

Mutire BN, Keraka MN, Kimuu PK, Kabiru EW, Ombeka VO, Oguya F. Factors associated with default from treatment among tuberculosis patients in Nairobi province, Kenya: a case control study. BMC Public Health [Internet]. 2011 Sep 9 321 [cited 2019 Jun 26];11(1):696. Available from: 322 <http://bmcpublihealth.biomedcentral.com/articles/10.1186/1471-2458-11-696>

Please check on the author guidelines page whether all detail in the last two lines above is necessary.

Reviewer: 2

Dr. Annika Sweetland, Columbia College of Physicians and Surgeons

Comments to the Author:

Major revision

This manuscript makes a unique contribution to the literature in that it is the first to examine depression trajectories in TB patients in the absence of mental health treatment. While I am not qualified to assess the statistical methods, if valid, the findings are of potential interest and use in the field. The authors describe several depressive symptom trajectories (declining, growth, or high), and effectively pinpoint one trajectory as being the most strongly associated with LTFU ("growth"). If true, this seems to suggest that ongoing monitoring of depressive symptoms in TB patients to identify those whose symptoms increase after baseline may help to identify this high-risk group for targeted intervention.

1. However, the authors do not explore this clinical implication. The baseline prevalence of depressive symptoms is also unclear, making the entire picture difficult to interpret. In order to strengthen this paper, I would want to have a clearer picture of the following:

a. what proportion of patients met criteria for "probable depression" at baseline (i.e. prevalence)?

This information is briefly added to the results section but is better detailed in a previous publication, which is also referenced.

b. what proportion of patients developed probable depression during the course of treatment (incidence)?

The incidence of individuals developing symptoms suggesting major depressive episodes during TB treatment was of 38.7%. This information is now included in the manuscript.

c. Of those with baseline depression, what proportion remitted without mental health treatment? What does this mean for clinical management?

As we do not have information on what type of treatment individuals received for depressive symptoms, or if they received any specific treatment at all, we cannot estimate that information. We are including this in the limitations of the study (discussion section).

d. Did all individuals in the "growth" trajectory meet criteria for probable depression at baseline?

No, 52% did, this information is now included in the manuscript after we describe which model was found.

e. How do you interpret that "high" was not as strong a predictor of LTFU as "growth" (assuming I am understanding this finding correctly)?

We do not think we have enough sample size in order to make a good analysis of this category, as we state in the discussion section. Both for the estimation of trajectory parameters and for the logistic regression the confidence interval does not allow for any certainty.

f. Overall, what are the clinical implications of these findings regarding TB and depression service integration?

The practical implications of our results has now been added in the discussion section.

Biostatistician review methods.

2. Minor: sample size should be reported in the abstract and methods section; fewer abbreviations would make the paper easier to read

a. The sample size is now reported in both the abstract and the methods section.

b. Some abbreviations (MDE, FH, SH) were replaced by plain text as we agree with the reviewer that it makes the manuscript easier to read. We kept some abbreviations that are frequently used such as TB and LTFU.

Reviewer: 3

Dr. Conrad Kabali, Health Quality Ontario

Comments to the Author:

1. I kindly suggest revising the terminology in page 12, line 35 to "95% Confidence Interval (CI)" instead of "IC95%".

We have replaced all instances of "IC95%" with "95%CI".

2. On line 42, it would be beneficial to consider the magnitude of the point estimate (OR=1.28) in addition to the tests of significance when interpreting the results.

We have amended the paragraph in order to note that a small but non-significant association was found.

3. In regards to Table 3, it may be appropriate to reconsider the inference made on covariates that serve only as potential confounders. For instance, if evaluating the effect of alcohol, it may be necessary to remove covariates that are in the causal pathway.

This information has been corrected in both results and discussion sections

4. Finally, on page 14, line 5, it is recommended to focus on the effect size rather than just the significance level, as the point estimate may not be precise enough to form a conclusion. Further data collection may be needed to make a conclusion.

The paragraph of the Discussion section to which this comment refers has been re-written in order to make it more readable. We have been more cautious in the wording of both the discussion and conclusion sections.

Reviewer: 4

Dr. Denis Talbot, Université Laval

Comments to the Author:

This is a statistical review of the paper entitled "Trajectories of depressive symptoms during the first half of drug-sensitive tuberculosis treatment and loss to follow up in Lima, Peru".

This paper is concerned with estimating the association between the trajectories of depressive symptoms during the first half of tuberculosis treatment with the probability of being "lost to follow up" during the second half of the treatment. I found the paper to read well and to provide a good amount of detail. I suggest some relatively minor revisions to both methods and reporting. While I focus on statistical aspects of the work in my paper, I wonder about the relevance of this study considering the small sample size and the ensuing imprecision of statistical results - which the authors appropriately recognized as a limitation of their study.

Specific comments/questions/suggestions:

1. Abstract: The problem under study wasn't very clear to me when reading the abstract. It wasn't clear to me that "loss to follow-up" was relative to patients' treatment and not relative to their participation in a longitudinal study. While I appreciate the challenges of space constraint, I recommend the authors revise the abstract to clarify.

The abstract has been rewritten in order to make the aim of the study clearer and adapt it to the journal's format.

2. Introduction:

a. What is XDR-TB?

XDR-TB stands for "Extensively drug-resistant tuberculosis" and it is now defined when it first appears in the manuscript.

b. I think the authors could strengthen the rationale for their study. What is the importance of knowing the association between depressive symptom trajectories and loss to follow-up? Can this be used by clinicians or public health to prevent loss to follow-up? Is this information relevant for future research? We have tried to improve and clarify the rationale for our study, it can be found in the last paragraph of the Introduction section.

3. Methods:

a. The exclusion of individuals with less than four measurements of depressive symptoms and those who developed MDR-TB during the follow-up may induce selection bias. If deemed possible and relevant, the authors may consider controlling this potential selection bias using inverse probability of censoring weighting or multiple imputation; the former being more easily applied than the latter with latent growth class analysis.

The reason to exclude those with less than 4 measurements of depressive symptoms was that they did not have the complete information to contribute to the estimation of the trajectories. However, it is true that this might induce some selection bias, and thus we are including some analysis of the characteristics of individuals that did not provide enough information for the estimation of trajectories.

b. CFI should be comparative fit index instead of comparative fix index.

We have taken into account the suggestion in the relevant section.

c. The authors may consider using the Guidelines for Reporting Trajectory Studies (GRoLTS) in addition to the STROBE to ensure that all appropriate information for this type of analysis is reported transparently. Some of the missing information for the methods include: Number of random start values, total number of models fitted including a one-class solution.

We appreciate the suggestion for the use of this instrument, the requested information is now included in the methods and results section of the manuscript.

d. Please clarify when the covariates that were used for adjustment were measured.

Covariates were measured at the start of follow-up, this is now included in the methods section.

e. Were there any missing data on the covariates? If yes, please describe in the results these missing data and clarify in the methods how they were treated.

We only included observations with complete data in the multiple logistic regression. The information on the number of missing values was added in the text describing the variables on the Results section.

f. Please provide the code for performing the trajectory analysis as supplementary material as recommended by GRoLTS.

We have added the code for the analysis as supplementary data.

4. Results:

a. I suggest adding a comparison of subjects included with those excluded to better inform on the potential for selection bias due to these exclusions.

In the results section of the manuscript, we have added a section in which we detail key characteristics of the excluded information.

b. Please remove the trajectories from table 1 since it is the result of an analysis and not a descriptive variable that can be directly measured.

We have removed the information on trajectories from table 1 as suggested by the reviewer.

c. Please remove the IC 95% from the legend of table 1 since this abbreviation is not used in the table.

The 95%CI legend is now removed from Table 1.

d. I suggest revising the interpretation of non-significant results as being inconclusive rather than as an absence of association. With very large confidence intervals, the data really provide no meaningful information on the presence, absence or strength of a potential association.

This information has been corrected in both results and discussion sections

e. If the covariates are considered as potential confounders, then their coefficients should not be interpreted and their reporting in Table 2 may be misleading (subject to Table 2 fallacy, see Table 2 Fallacy: Presenting and Interpreting Confounder and Modifier Coefficients | American Journal of Epidemiology | Oxford Academic).

We appreciate the call on this issue as well as the literature provided. We include a warning sentence after the call to table 2 and have noted this in the discussion section, as readers should be aware of the potential for bias when interpreting the coefficient for potential confounders. However, we think it is still important to present the data in the interest of readers that might be looking for information on the association between our outcome variable and a covariate for which there is little research.

f. Please add a plot of the observed and predicted means for each trajectory group to visually assess the goodness of fit (see GRoLTS).

This graphic is now part of the supplemental material (supplemental material 3).

5. Discussion:

I find the paper to insufficiently describe the practical implications of their results. Given the lack of statistical power, perhaps there are no immediate implications?

The practical implication of our results has now been added in the discussion section.

6. In my opinion, a minor English language revision is needed.

We have performed a complete review of the new version of the manuscript in order to improve its readability and use of English.

VERSION 2 – REVIEW

REVIEWER	Sweetland, Annika Columbia University Vagelos College of Physicians and Surgeons, Psychiatry
REVIEW RETURNED	15-May-2023

GENERAL COMMENTS	Dear editor, This version is much improved, particularly with respect to the implications. There are still minor errors throughout, mostly grammatical, as described below. The only substantive question is related to the timing of the assessments. The timing is unclear about when baseline and three monthly follow ups were done. In the figure it looks like it was baseline and months 1-3 of treatment, in which case the patient would only be halfway through TB treatment. The results suggest that the last assessment was upon TB treatment completion. Were the assessments possibly baseline, 2, 4, and 6 months? Or were they any three follow up times (i.e. variable by individual)? This needs more clarity.
--

	Minor/grammatical: “TB was the first cause of death” should be changed to “leading” cause of death; same in next sentence with “second leading” Page 5, lines 89-92: this sentence is confusing. The authors appear to be citing their own previous work in which 38% and 14% prevalence was found at the beginning and end of treatment, but that reference (#12) is not cited. It would be clearer if it stated “In a previously published study among the same cohort, the prevalence of depression at baseline and 6 months was X and Y, respectively. Page 7, line 114 – replace “previous with “prior” Page 7, line 117, replace “populated” with “populous” Page 7 line 118, suggest deletion of “large migration” since “migrants” is already stated twice in the same sentence Page 10, line 189: missing “%” after 13.5 Missing word “treatment” in Title and Abstract: Are trajectories of depressive symptoms during the first half of drug-sensitive tuberculosis TREATMENT associated with loss to follow-up? 1 a, b, d, e f are adequately addressed; 1c is not –the limitation that no data is available on whether TB patients received any sort of depression intervention during treatment has been added but is not clear. Suggest rewording to “no information is available about whether participants received any depression intervention during TB treatment” . Abstract, results: not necessary to include the denominator (265) for the 3 trajectories, just number (%) Reference 14: this reference does not look correct or complete
--	---

REVIEWER	Kabali, Conrad Health Quality Ontario
REVIEW RETURNED	22-Apr-2023

GENERAL COMMENTS	Comments addressed. No further comments
---

REVIEWER	Talbot, Denis Université Laval
REVIEW RETURNED	04-May-2023

GENERAL COMMENTS	The revised manuscript has satisfactorily addressed most of my comments. I have a few remaining concerns:  - Selection bias should be mentioned as a limitation in this study given that missingness – and thus selection into the study – is associated with the outcome and likely also associated with the exposure of interest. - Concerning the interpretation of potential confounder coefficients, I disagree with the authors that including this information can be helpful to readers interested in the association between the covariates and the outcome, since this information can be misleading. At the very least, the “Table 2 Fallacy paper” should
--

	cited after their sentence cautioning readers about the interpretation of these coefficients. Minor: This sentence is very lengthy and has multiple “and”, making hard to follow. I suggest dividing into two or three sentences. “We included information on individuals that had at least four complete measurements of depressive symptoms (baseline and three-monthly follow-ups), and excluded those who were diagnosed with MDR-TB at any point during follow-up and those who died during the first 3 months of treatment and thus were not at risk during the second half of TB treatment and could not contribute to depressive measures in the first half of TB treatment.” The English language has been improved but some revision is still needed. Here are some suggested edits that I noted while reading the revised manuscript:  - Abstract Objective: pulmonary tuberculosis (PTB) instead of pulmonary TB (PTB) - Abstract Results: Three trajectories were found - Abstract Results: Compared to those with a "Declining" trajectory - Introduction: that ranged from 38% at the start of TB treatment to 14% at the end of it, while it was estimated to 7% - Methods: The number of random starts was of 500 for the initial stage, while 10 optimizations were used for the final stage. - Methods: This step provided us with the information on the trajectories (intercept, linear and quadratic terms) - Table 2: Change Lineal to Linear - Discussion: there has been no local evaluation regarding its performance when multiple measurements are taken over a relatively short period (6 months).
--	--

VERSION 2 – AUTHOR RESPONSE

Reviewer: 2

Comment: This version is much improved, particularly with respect to the implications. There are still minor errors throughout, mostly grammatical, as described below. The only substantive question is related to the timing of the assessments. The timing is unclear about when baseline and three monthly follow ups were done. In the figure it looks like it was baseline and months 1-3 of treatment, in which case the patient would only be halfway through TB treatment. The results suggest that the last assessment was upon TB treatment completion. Were the assessments possibly baseline, 2, 4, and 6 months? Or were they any three follow up times (i.e. variable by individual)? This needs more clarity.

Response: We have clarified the timing of the depression measurements, this now can be found in the “Study Variables” subtitle of the “Methods” section: “The main predictor variable was trajectories (patterns of change over time) of depressive symptoms during the first half of PTB treatment. These trajectories were identified as part of the data analysis of this study using the score of a 5-item, locally validated version of the CES-D in Spanish¹⁶. Participants completed this scale at the beginning of TB treatment and then monthly thereafter until treatment completion (6 months), LRFU, or the initiation of treatment for MDR-TB. Each participant contributed between 1 to 7 monthly scores of depressive symptoms.”.

Also, in the “Study Design and Population” subtitle of the “Methods” section (third paragraph), we modified the text as suggested by the reviewers: “For this analysis, we included information on

individuals that had at least four complete measurements of depressive symptoms (baseline and three-monthly follow-ups), and excluded those who were diagnosed with MDR-TB at any point during follow-up and those who died during the first 3 months of treatment and thus were not at risk during the second half of TB treatment and could not contribute to depressive measures in the first half of TB treatment”.

Comment:

Minor/grammatical:

- “TB was the first cause of death” should be changed to “leading” cause of death; same in next sentence with “second leading”
- Page 5, lines 89-92: this sentence is confusing. The authors appear to be citing their own previous work in which 38% and 14% prevalence was found at the beginning and end of treatment, but that reference (#12) is not cited. It would be clearer if it stated “In a previously published study among the same cohort, the prevalence of depression at baseline and 6 months was X and Y, respectively.

- Page 7, line 114 – replace “previous with “prior”
- Page 7, line 117, replace “populated” with “populous”
- Page 7 line 118, suggest deletion of “large migration” since “migrants” is already stated twice in the same sentence
- Page 10, line 189: missing “%” after 13.5
- Missing word “treatment” in Title and Abstract: Are trajectories of depressive symptoms during the first half of drug-sensitive tuberculosis TREATMENT associated with loss to follow-up?1
- 1 a, b, d, e f are adequately addressed; 1c is not –the limitation that no data is available on whether TB patients received any sort of depression intervention during treatment has been added but is not clear. Suggest rewording to “no information is available about whether participants received any depression intervention during TB treatment” .
- Abstract, results: not necessary to include the denominator (265) for the 3 trajectories, just number (%)
- Reference 14: this reference does not look correct or complete

Response: We appreciate the input regarding the wording in the English language, all changes have been addressed. Regarding the inclusion of the denominator, this was suggested by the Journal’s editors, but we are happy to adapt it as the journal requires.

Reviewer: 4

Comments to the Author:

The revised manuscript has satisfactorily addressed most of my comments. I have a few remaining concerns:

Comment: Selection bias should be mentioned as a limitation in this study given that missingness. Concerning the interpretation of potential confounder coefficients, I disagree with the authors that including this information can be helpful to readers interested in the association between the covariates and the outcome, since this information can be misleading. At the very least, the “Table 2 Fallacy paper” should be cited after their sentence cautioning readers about the interpretation of these coefficients.– and thus selection into the study – is associated with the outcome and likely also associated with the exposure of interest.

We have added the possibility of selection bias in the limitations section of the manuscript. Regarding the second point, we have made the readers aware that the interpretation of the coefficients in Table 3 might be misleading and needs to take into account the possibility that the adjusted coefficients of covariates on a multiple regression model might not be properly adjusted even if the main response variable is, we cite the suggested literature as well as another paper we found interesting, as a further example.

The last paragraph of the discussion section now reads as: “Readers should also be aware of the possibility of selection bias influencing the results, as LTFU and possibly the trajectories of depressive disorders may be associated with a higher likelihood of missing data. Finally, caution should be exercised when interpreting adjusted coefficients of the multiple logistic regression model, as even though the coefficient of the main variable may be properly adjusted, the coefficients of covariates may not be, as discussed by Westreich and Greenland and exemplified by Gretchen et al.”

Comment: This sentence is very lengthy and has multiple “and”, making it hard to follow. I suggest dividing it into two or three sentences. “We included information on individuals that had at least four complete measurements of depressive symptoms (baseline and three-monthly follow-ups), and excluded those who were diagnosed with MDR-TB at any point during follow-up and those who died during the first 3 months of treatment and thus were not at risk during the second half of TB treatment and could not contribute to depressive measures in the first half of TB treatment.”

Response: The suggested change has been implemented, now it reads: “For this analysis, we included information on individuals that had at least four complete measurements of depressive symptoms (baseline and three-monthly follow-ups), and excluded those who were diagnosed with MDR-TB at any point during follow-up and those who died during the first 3 months of treatment and thus were not at risk during the second half of TB treatment and could not contribute to depressive measures in the first half of TB treatment.”

Comment: The English language has been improved but some revision is still needed. Here are some suggested edits that I noted while reading the revised manuscript:

- Abstract Objective: pulmonary tuberculosis (PTB) instead of pulmonary TB (PTB)
- Abstract Results: Three trajectories were found
- Abstract Results: Compared to those with a "Declining" trajectory
- Introduction: that ranged from 38% at the start of TB treatment to 14% at the end of it, while it was estimated to 7%
- Methods: The number of random starts was of 500 for the initial stage, while 10 optimizations were used for the final stage.
- Methods: This step provided us with the information on the trajectories (intercept, linear and quadratic terms)
- Table 2: Change Lineal to Linear
- Discussion: there has been no local evaluation regarding its performance when multiple measurements are taken over a relatively short period (6 months).

Response: We appreciate the feedback, the corrections were implemented.

VERSION 3 – REVIEW

REVIEWER	Talbot, Denis Université Laval
REVIEW RETURNED	20-Jun-2023

GENERAL COMMENTS	All my previous comments have been addressed. Well done! I noticed two minor typos when reviewing this version: - Strengths and limitations: The [limited] sample size was limited, ... - Study variables: Participants completed this scale at the beginning of TB treatment and then monthly thereafter until treatment completion (6 months), [LRFU] LTFU, ...
--